# The Performance of Dual-Task Tests Can Be a Combined Neuro-Psychological and Motor Marker of Mild Cognitive Impairment, Depression and Dementia in Geriatric Patients—A Cross-Sectional Study

**DOI:** 10.3390/jcm10225358

**Published:** 2021-11-17

**Authors:** Agnieszka Kasiukiewicz, Lukasz Magnuszewski, Marta Swietek, Zyta Beata Wojszel

**Affiliations:** 1Department of Geriatrics, Medical University of Bialystok, 15-471 Bialystok, Poland; wojszel@umb.edu.pl; 2Department of Geriatrics, Hospital of the Ministry of Interior and Administration in Bialystok, 15-471 Bialystok, Poland; magnuszewskilukas@gmail.com (L.M.); marta.swietek@umb.edu.pl (M.S.); 3Doctoral Studies, Department of Geriatrics, Faculty of Health Sciences, Medical University of Bialystok, 15-471 Bialystok, Poland

**Keywords:** dementia, cognitive decline, biomarkers, dual-task, gait, geriatric patients

## Abstract

The study aims to assess the performance of dual-task tests in the geriatric population and their association with the cognitive status of the patients. Methods: Patients admitted to the Department of Geriatrics, Hospital of the Ministry of Interior and Administration on Bialystok, Poland, in 2019 and 2020 were enrolled in the study. Data on the patients’ clinical, functional, and cognitive status were collected based on the comprehensive geriatric assessment. Dual-task tests included Timed Up and Go (TUG) test while counting backward (CB7), enumerating animals (EA), and holding a cup (TUG M). Results: 250 patients were included in the study, with a median age of 81.5 years (IQR 76–86) and most above 75 years of age (80.8%). Only 29 (11.6%) of study participants had no cognitive or mood disorders. Depression was diagnosed in 30.4%, MCI in 12%, and dementia in 38.4% of cases with median Mini-Mental Score Evaluation (MMSE) 17 (12–20) points. Dual-task TUG CB7 results did not differ between cognitive conditions of patients. TUG EA differed between healthy controls and other cognitive groups and TUG between healthy controls and depression and dementia, but not mild cognitive impairment (MCI). The performance of all dual-task tests differed in patients with and without dementia. Ability to finish TUG CB7 was low even in the group without dementia. There were statistically significant differences in median scores of MMSE and Clock Drawing Test (CDT) between patients who were able or not to finish single and dual-task gait tests. Conclusion: Dual-task test results and the performance of these tasks can differentiate patients with depression, MCI and dementia compared to healthy controls in the geriatric population.

## 1. Introduction

Dementia and cognitive decline are global issues, as they often lead to functional disability and dependency of older people [1]. They burden not only patients but also family caregivers and, in addition, the medical system and economic resources. As the community ages, the problem of dementia rises. Therefore, it is crucial to diagnose cognitive decline in the early stages to implement pharmacological and non-pharmacological treatment to slow down the process and prevent disability.

However, analyses show that dementia is still an underdiagnosed condition [2]. It is due to several reasons, including a limited number of specialists (geriatricians, neuropsychologists) specializing in older patients’ disorders, lack of established biomarkers of early stages of decline and “insensitivity” of society, and tendency to call memory decline as typical to aging.

Gait has generally been considered an automatic process, not involving cognitive functions. However, the research emphasizes the role of cognitive function as a component of motor control in the elderly [3]. Lundin-Olsson et al., proved that dual-task (talking while walking paradigm) was an accurate predictor of falls. Participants, who stopped during the cognitive task, i.e., speaking, had a higher risk of falling [4].

Executive functions play a crucial role in regulating gait when new activities are undertaken or modified previously learned movement schemes. Neuroimaging studies confirmed a common pattern of neural activity for pace, executive function, and attention involving the frontal cortex and its cortical-subcortical neural network [5]. In addition, many studies have shown that decreased executive function can contribute to the development of gait disturbances and increase the risk of falls [6]. Dual-task gait tests measure the cognitive reserve capacity of the brain while the simultaneous performance of two—cognitive and motor-tasks [7]. They include straight-line walking or a timed “up and go” test, while performing another attention-demanding task, like counting backward numbers, months, or enumerating animals. Thus, they can be helpful as a marker of low cognitive reserve capacity and cognitive decline. However, the limitation of these tests’ usage is the lack of consensus on which dual-task to use and the most predictable measurements. Additionally, a limited number of studies include the geriatric population, burdened with functional disability and multimorbidity.

Our study aims to assess the performance of dual-task tests in the geriatric population and their association with the cognitive status of the patients.

## 2. Materials and Methods

All consecutive patients admitted to the Department of Geriatrics, Hospital of the Ministry of Interior and Administration in Bialystok, Poland at the turn of 2019 and 2020, were enrolled in the study. The Department of Geriatrics is a subacute ward. It admits patients over 60 years old, mainly with multimorbidity and functional decline. A geriatric team (consisting of geriatricians, geriatric nurses, physiotherapists, and psychologists) provides a comprehensive geriatric assessment and creates a long-term care plan. We assumed, following epidemiological data, that 10% of the elderly population has dementia and the other 10% have a mild cognitive impairment, and we adopted a 5% margin of error and 95% confidence interval. The calculated sample size was 246.

### 2.1. Patient Characteristics

Based on the medical interview (caregivers verified information obtained from the patients), physical examination, tests, and measurements used during the comprehensive geriatric assessment, the following data were collected:-sociodemographic—age, gender, education, place of residence (urban/rural),-clinical—weight, height, BMI, chronic diseases (peripheral arterial disease, ischemic heart disease, chronic heart failure, history of myocardial infarction, hypertension, atrial fibrillation, history of transient ischemic attack (TIA) or stroke, chronic obstructive pulmonary disease, diabetes, neoplasm, dementia, parkinsonism, chronic osteoarthritis, osteoporosis, and chronic renal disease), Charlson Comorbidity Index, medicines taken before hospitalization,-functional efficiency—the ability to perform basic activities of daily living was assessed with the Barthel index [8], and the ability to perform instrumental activities of daily living (IADL) was assessed with six items of the Duke Older American Resources and Services (OARS) I-ADL [9].-frailty score—assessed with FRAIL scale [10] patients were scored as frail if they met three or more out of 5 criteria.

### 2.2. Cognitive Function

A geriatrician scored patients with Mini-Mental State Examination (MMSE) [11], 7-point Clock Drawing Test (CDT) [12], and 15-item Geriatric Depression Scale (GDS) [13]. A neuropsychologist confirmed the diagnosis of dementia, mild cognitive impairment, or depression based on a more thorough examination. We defined the cognitively intact patients (normal mental state) as having MMSE > 26 points and no depression or other psychiatric disorders.

### 2.3. Gait Measurements

The trained physiotherapist performed single task measurement with a Timed Up-and-Go test (TUG) [14]. Patients received instructions before taking a test. The test consists of the following tasks:Rising from the chair;Walking 3 m with usual gait speed;Turning around;Stepping back and sitting down in the chair.

Measurements included time of the performance of the tasks, stride number during the test, stride number during turning around; furthermore, they were additionally normalized, meaning division by body height.

Dual-task measurements consisted of 3 tests:-TUG test with simultaneously counting backward from 100 by 7 (TUG CB7) [15];-TUG test with simultaneously enumerating different animals (TUG EA) [15];-TUG test with simultaneously holding a cup filled with water (TUG manual) [16].

Measurements included similar parameters as in single task TUG. In addition, we computed dual-task cost (DTC, %) that expresses the reduction in performance when making additional tasks as (TUG dual-task time—TUG single-task time)/TUG single-task time) x100%. According to the previous literature, we defined DTC > 20% as a significantly increased risk of progressing to dementia [17].

We determined the following performance status:-normal—patients able to finish the test without problems with gait performance or cognitive task performance;-not performed due to cognitive problems—not able to perform correctly cognitive task before TUG or not able to understand the instruction or mistakes during performing cognitive task (3 mistakes in counting backwards, enumerating less than 3 animals) or patient stops when performing cognitive task;-Not performed due to physical problems—not able to perform gait task.

### 2.4. Statistical Analysis

STATISTICA 13.3 software package (TIBCO Software, Palo Alto, CA, USA) was used to analyze the data collected. The Shapiro–Wilk test was used to assess the distribution of variables. Data were presented as medians (Me) and interquartile range (IQR) and the number of cases and categorical variables’ percentages. Proportions were compared using χ^2^ tests or Fisher’s exact test, as appropriate, while the Mann–Whitney U test and the Kruskal–Wallis test by rank were used to compare the distribution of continuous variables. Differences between multiple subgroups were compared with the Bonferroni post-hoc test. The strength of association was measured using Spearman’s rank correlation coefficient. Missing values were omitted, and statistics in such cases were calculated for the adequately reduced groups. A *p*-value of less than 0.05 was regarded as significant.

### 2.5. Ethics Approval

The Ethics Committee approved the study at the Medical University of Bialystok. All procedures performed in the study were under the ethical standards of the Medical University of Bialystok research committee and with the Helsinki declaration. All study participants gave their informed consent to participate in it.

## 3. Results

### 3.1. Study Cohort Characteristics

A total of 250 patients were hospitalized in the Department during the study period and included. The characteristics of the study group with subgroups of patients with or without dementia are presented in Table 1. The median age of the patients was 81.5 years (IQR 76–86), and most of them were above 75 years of age (80.8%), female (73.2%), and living in the urban community (88.4%). They were burdened with disability in physical (P-ADL: Barhel Index Score median 85 (65–95) points) and instrumental activities of daily living (IADL—median 6 (3–10) points) and frailty (FRAIL score Me 3 (2–3)). In addition, they were multimorbid, with Charlson Comorbidity Index 6 (4–7) and high occurrence of chronic arthritis (65.6%), heart failure (39.2%), diabetes (48.4%), Parkinson’s syndrome (12.4%). Dementia +/− subgroups did not differ in gender, place of residence, multimorbidity, and chronic conditions. Patients with dementia were older (*p* = 0.001) and more functionally disabled in both PADL and IADL (*p* < 0.001).

### 3.2. Mental State of the Patients

Only 29 (11.6%) study participants had no cognitive or mood disorders (healthy controls). Depression was diagnosed in 76 (30.4%) and mild cognitive impairment in 30 (12%) of patients. Dementia was diagnosed in 38.4% of cases with median MMSE 17 (12–20) points and CDT 0 (0–3) points. A total of 44 (17.6%) patients were diagnosed with mild stage, 30 (12%) with moderate, and 22 (8.8%) with a severe stage of dementia. The cognitive state was also altered due to other psychiatric disorders (schizophrenia, mental disability) in 8 patients and due to the general condition (thyroid dysfunction, anemia, infection) in 11 patients.

### 3.3. Single Task and Dual-Task Gait Test Results

Single task Timed Up-and-Go test was performed by 211 (84.4%) of patients. Eight (3.2%) patients did not finish the test due to problems with understanding instruction and 11 (4.4%) due to physical problems. The median time of the TUG test was 17 s (12.7–26.5), stride number 24 (18.5–31), and stride number during turning around—6 (5–8).

Dual-task TUG CB7, while counting backward for 7, was able to perform only in 36 (14.4%) cases. Patients mainly had problems with cognitive tasks or understanding instruction (70.8%); in 14.8%, they did not finish the test due to physical problems. Participants who completed the test scored on average 16 (12–22.5) seconds, with 23 (18–27.5) steps during the trial and 6 (5–7) steps while turning around. Dual-task cost was 34 (22–61)%.

TUG with enumerating different animals (TUG EA) was performed by 128 (51.2%) patients and not finished in 81 (32.4%) cases due to cognitive problems and in 41 (16.4%) physical problems. Median time scored 18 (14–26) seconds, with 23.5 (19.5–29.5) steps and 6 (5–8) while turning around. DTC was 19 (0–41)%.

A total of 155 (62%) cases TUG manual; 23 (9.2%) had cognitive and 72 (28.2%) physical problems while performing tasks. The median score was 16 (13–23) seconds, stride number during the test—23 (19–28), and while turning around—6 (5–8). DTC was 11 (0–25)%.

### 3.4. Dual Task Test Scores and Cognitive State of Patients

When comparing patients with and without dementia, only scores in single task TUG (time, stride number, and stride number while turning around) were significantly different (*p* < 0.01); in dual-task tests, it concerned only stride number while turning around (*p* = 0.01).

For more detailed analysis, we excluded patients with alterations in cognitive tests due to other psychiatric disorders and the general condition (19 cases). We divided patients into subgroups with normal mental state, depression, MCI, and three stages of dementia. Results are presented in Table 2.

Single task TUG time differed between healthy controls and moderate and severe stages of dementia. Differences between depression and dementia were observed in the severe stage of dementia. There were no statistically significant differences between MCI, depression, and dementia (mild, moderate) in test time. Depression and MCI differed in stride number during turning around.

Dual-task TUG CB7 measurements were not statistically significant between particular subgroups in any parameter. TUG EA differed between healthy controls and depression at the time of the test. When counting strides and stride numbers while turning around, there were differences between healthy controls and MCI and dementia but not depression. TUG manual differed in the test time between healthy controls and depression and dementia, but not MCI. Differences between depression and dementia patients were visible but in moderate and severe stages of dementia. There were no statistically significant differences between subgroups in dual task cost in any dual task TUG. Normalized test results (the result divided by body height) also did not differ between groups.

### 3.5. Performance of Dual-Task Gait Tests and Cognitive State of Patients

When looking at patients’ cognitive state (Table 3), single task TUG performance was significantly lower only in the severe stage of dementia. TUG CB7 was the worst; no patient with severe dementia and only one patient with moderate dementia was able to finish the test. Performance was also different between patients with normal cognitive state and other subgroups. TUG EA was performed by a lower number of patients with dementia than healthy controls; additionally, there were differences between depression and moderate dementia. No patient with severe dementia was able to finish the test. TUG manual performance differed between healthy controls and patients with depression and dementia.

In Table 3, we compared average scores of cognitive tests in subgroups of patients with the different performance of dual-task tests. There were statistically significant differences in median scores of MMSE and CDT between patients who were able or not to finish single and dual-task gait tests.

### 3.6. Correlation between Cognitive Test Scores and Dual-Task Tests Measurements

The strength of association between MMSE score and dual-task tests is presented in Table 4. Statistical analysis using Spearman’s rank correlation coefficient showed a significant association MMSE score with all measurements of TUG while naming animals and manual, but not while counting backward, even after taking into analysis normalized measures. Similar results were found for the CDT score. However, Dual-Task Cost correlation with cognitive tests was not statistically significant.

## 4. Discussion

This study aimed to evaluate changes in gait test performance in patients of different cognitive statuses. The slowness of gait speed is a well-proven factor associated with cognitive impairment and a risk factor for dementia, especially in frail patients [18]. In recent years an increasing interest in dual-task interference has been evaluated. In a systematic review provided by Mancioppi G et al., 1939 records were identified through database searching years 2010–2020, with 347 full-text articles and 38 studies included in the meta-analysis [19]. The use of single and dual gait speeds was recommended in Canadian guidelines on non-cognitive markers of dementia [17].

Our results are consistent with other analyses, which show the association of dual-task tests with cognitive function [20,21,22]. Participants walked the standard TUG for the dual-task animals while subtracting serial 7s from 100 aloud or enumerating animals aloud. These two different dual-task trials were selected based on previous research, demonstrating that subtractions depend more on working memory and attention. At the same time, enumerating animals aloud is more related to verbal fluency, which relies on semantic memory. While performing the dual-task trials, there was no instruction to prioritize the gait or cognitive task. Allowing both walking and cognitive tasks have previously been shown to provide a more realistic representation of the daily living activities of older individuals [23,24].

However, when taking into account only dementia or non-dementia patients, this correlation may not be visible, as these patients are diverse. For this reason, we have decided to detail the mental condition of patients: healthy controls, depression, MCI, and three stages of dementia. It was also important to exclude patients who could have worse results of cognitive tests due to other psychiatric disorders, like schizophrenia or poor general state. Otherwise, it may cause reversible cognitive decline (infection, hypoglycemia, electrolyte imbalance, anemia, or thyroid diseases). Our results show that single and dual-task gait parameters can differentiate healthy controls from patients with MCI and dementia (TUG, TUG NA) or depression (TUG manual). A similar effect was found in research by Ahman HB et al., where dual-task tests with enumerating animals and counting months backward discriminated between dementia, mild cognitive impairment, subjective cognitive impairment, and healthy patients [25]. The effect of TUG while counting back from seven in our analysis was not significant. However, this impact could have been decreased by the low number of cases (patients who could finish the test). Similarly, differentiation between MCI and dementia or stages of dementia was not significant in analyses, possibly because of the low number of patients with moderate and severe dementia who finished dual-task tests.

The correlation between cognitive test scores (MMSE, CDT) and dual-task test measurements was significant. However, the significance was weak. Walking test results are often affected by other neurologic disorders, arthritis, or general health conditions in the geriatric population. We have decided to evaluate depression and treat it separately from healthy controls as it is a proven risk factor that can occur before the onset of dementia [26]. The differences were visible in single TUG and slowness of gait speed, the manual TUG, and the percentage of the performance of dual-task tests. To our knowledge, there are not many studies that evaluate mood disorders in the context of gait speed and dual-task as decreased cognitive reserve capacity. These published ones seem to confirm the worse performance of gait tests by patients with depression than healthy persons but better than MCI patients [27].

The strength of our study is the evaluation of different cognitive and mood states and the assessed study population, which consists of typical geriatric patients with a median age of 81.5 years. Other studies that investigated dual-task and cognition included younger patients with an average age of 65–70 years [25,28,29]. As in our study, older people are multimorbid and functionally disabled, which can impact the practical use of gait diagnostic tests. Dual tasks were previously evaluated in patients in different stages of activities of daily living (ADL). There were differences among groups during cognitive-motor tests, but not the motor and cognitive tests separately [30]. In a large percentage, our patients were not able to finish dual-task tests, especially while counting backward for 7. It is worth paying attention that inability to complete dual tasks was also observed in patients without cognitive disorders, who could perform single task TUG. A higher percentage of performance was observed in NA and manual TUG. These are findings not investigated in previously published studies. Our analyses prove that the performance rate can be a marker of cognitive decline associated with MMSE and CDT scores. Patients without dementia, who did not finish dual-task tests, had the longer time of single task TUG (19 s vs. 11.2 s for performance TUG CB7; 20.1 s vs. 15 s for TUG NA and 33.7 s vs. 15 s for TUG M; *p* < 0.05—data not presented in results). As walking speed can be a marker of the low reserve capacity of the brain, during dual tasks, these reserves seem to deplete.

Our study has some limitations: it was a cross-sectional analysis, while prospective evaluation could answer whether low dual-task scores can predict cognitive decline. We did not measure some parameters which were proven to be significant in other analyses, as the number of words/10 s or enumerating months backward. However, a recent meta-analysis shows that studies use different measurements and give different results [31]. There is no consensus so far, which tests and parameters should be used to discriminate cognitive impairment. We also did not differentiate the types of dementia and particular cognitive functions, including executive functions with standardized instruments (as Trail Making Tests or Stroop test) that could significantly impact the significance of specific tests. Since the TUG trial was not possible in a large part of geriatric ward patients, it also limits the possibility of generalizing the results to the entire population of older adults. Nevertheless, this analysis shows that DT tests can be a helpful tool in assessing cognitive functions used by physiotherapists or nurses during the comprehensive geriatric assessment. In addition, they can help geriatricians and other specialists diagnose early stages of mental disorders.

## 5. Conclusions

Dual-task gait tests are a promising tool used during comprehensive geriatric assessments by physiotherapists or nurses to differentiate patients with depression, MCI, and dementia from healthy controls in the senior population. Not only gait parameters, but also the ability to perform dual tasks, can be a marker of the early stage of cognitive decline in functionally disabled patients.

## Figures and Tables

**Table 1 jcm-10-05358-t001:** Characteristics of the study population.

Parameter	Total	Dementia+ Group	Dementia− Group	*p* Value ^a^	Missing Values
No. (%) of patients	250	96 (38.4)	154 (61.6)		-
Age, y, M (SD)	81.5 (76.0–86.0)	83 (78.0–88.0)	80.5 (75.0–85.0)	0.001	-
Age, 75+ years, *n* (%)	202 (80.8)	86 (89.6)	116 (75.3)	0.005	-
Sex, women, *n* (%)	183 (73.2)	68 (70.8)	115 (74.7)	0.5	-
Place of residence, rural, *n* (%)	29 (11.6)	12 (12.5)	17 (11.1)	0.89	-
Barthel Index, Me (IQR)	85 (65–95)	75 (50–90)	90 (75–95)	<0.001	
IADL, Me (IQR)	6 (3–10)	3 (0–6.5)	8 (5–10)	<0.001	
Charlson Index, Me (IQR)	6 (4–7)	6 (4–8)	5.5 (4–7)	0.1	-
MMSE, Me (IQR)	23 (17–27)	17 (12–20)	26 (24–28)	<0.001	24
CDT, Me (IQR)	4 (0–6)	0 (0–3)	5 (3–6)	<0.001	27
Heart failure, Yes, *n* (%)	98 (39.2)	37 (38.5)	61 (39.6)	0.87	-
Diabetes, Yes, *n* (%)	121 (48.40)	43 (44.8)	78 (50.7)	0.37	-
Orthostatic hypotension, Yes, *n* (%)	85 (34.0)	35 (36.5)	50 (32.5)	0.52	-
Arthritis, Yes, *n* (%)	164 (65.60)	59 (61.5)	105 (68.2)	0.28	-
Parkinson syndrome, Yes, *n* (%)	31 (12.40)	13 (13.5)	18 (11.7)	0.67	-
History of stroke, Yes, *n* (%)	52 (20.8)	22 (22.9)	30 (19.5)	0.52	-
Ischemic heart disease, *n* (%)	80 (32.0)	30 (31.3)	50 (32.5)	0.84	-
Asthma/COPD, Yes, *n* (%)	39 (15.6)	10 (10.4)	29 (18.8)	0.07	
FRAIL Score, Me (IQR)	3.0 (2.0–3.0)	3 (2–4)	3 (2–3)	0.06	7

^a^—χ^2^ test or Fisher exact test, as appropriate, for categorical variables; Mann–Whitney test for interval variables; Abbreviations: IQR—interquartile range; Me—median value; *n*—number of cases; MMSE—Mini Mental State Examination, CDT—Clock Drawing Test; COPD, chronic obstructive pulmonary disease.

**Table 2 jcm-10-05358-t002:** Dual task gait test results in different cognitive states of patients.

Parameter	Normal (*n* = 29)	Depression (*n* = 76)	MCI (*n* = 30)	Mild Dementia (*n* = 44)	Moderate Dementia (*n* = 30)	Severe Dementia (*n* = 22)
Single task TUG						
TUG, s, Me (IQR)	12 (10.7–16)	18.4 (13–25.9)	16 (12–26)	17 (13–28)	23.7 (16–31) *	23.5 (17–43) *
Stride, *n*, Me (IQR)	19 (17–23)	23 (19–30)	25 (17–31)	25 (20–32) *	30 (22–35)	29.5 (21.5–40) *
Turning around, stride, Me (IQR)	4 (4–6)	6 (5–8) **	6 (5–8) *	6 (5–8) *	6 (5–9) *	6.5 (6–11.5) *^,^ **
Performance, yes, *n* (%)	27 (93.1)	69 (90.8)	29 (96.7)	37 (84.1)	23 (76.7)	12 (54.5) *^,^ **^,^ ***
Not performed, cognitive probl, *n* (%)	-	1 (1.3)	-	1 (2.27)	2 (6.7)	4 (18.2)
Not performed, physical probl, *n* (%)	2 (6.9)	6 (7.9)	1 (3.3)	6 (13.6)	5 (16.7)	6 (27.3)
TUG—CB7						
TUG, s, Me (IQR)	14.5 (12–16)	16 (11–29)	19.5 (13–25)	18.5 (13.5–22.5)	23	-
Stride, *n*, Me (IQR)	19 (18–25)	23 (14–28)	27.5 (19.5–32.5)	22.5 (18.5–26)	41	-
Turning around, stride, Me (IQR)	6 (5–7)	6 (5–7)	7 (6–8.5)	5.5 (6–6)	8	-
Performance, yes, *n* (%)	10 (34.5)	15 (19.7)	4 (13.3)	4 (9.1) *	1 (3.3) *	0 *^,^ **
Not performed, cognitive probl, *n* (%)	17 (58.6)	51 (67.1)	23 (76.7)	33 (75)	24 (80)	-
Not performed, physical probl, *n* (%)	2 (6.9)	10 (13.2)	3 (10)	7 (15.9)	5 (16.7)	16 (72.7)
Dual task cost, Me (IQR)	32.1 (27.7–36.3)	36.7 (16.7–79.3)	48.8 (18.8–61.3)	22.8 (14.1–78.3)	35.3	6 (27.3)
Cost >20%, *n* (%)	8 (80)	11 (73.3)	3 (75)	3 (75)	1 (100)	-
TUG—EA						
TUG, s, Me (IQR)	15 (12–18)	22 (13–30) *	18.5 (15–22)	18 (15–26)	25 (18–30)	-
Stride, *n*, Me (IQR)	20 (17–24)	22 (19–31)	26 (21–30)	23.5 (18–26)	32.5 (25–42) *^,^ **	-
Turning around, stride, Me (IQR)	6 (5–6)	6 (5–8)	7 (6–9) *	6 (5–8) *	9 (7–11) *^,^ **	-
Performance, yes, *n* (%)	23 (79.3)	51 (67.1)	18 (60)	22 (50) *	6 (20) *^,^ **	- *^,^ **^,^ ***
Not performed, cognitive probl, *n* (%)	4 (13.8)	15 (19.7)	9 (30)	15 (34.1)	16 (53.3)	16 (72.7)
Not performed, physical probl, *n* (%)	2 (6.9)	10 (13.2)	3 (10)	7 (15.9)	8 (26.7)	6 (27.3)
Dual task cost, Me (IQR)	18.2 (5.9–33.3)	11.7 (−6,7–50.1)	18.8 (0–25)	24 (−2.5–66.7)	25.8 (5.9–56.3)	-
Cost >20%, *n* (%)	11 (47.8)	20 (39.2)	8 (44.4)	13 (59.1)	4 (66.7)	-
TUG—manual						
TUG, s, Me (IQR)	13.5 (11–16)	18 (14–26) *	16 (14–21.8)	17 (13–20) *	22 (16–32.5) *	22 (15–22)
Stride, *n*, Me (IQR)	20.5 (18–25)	22 (18–28)	23.5 (20–28.5)	23 (19–26)	31 (24–35.5) *^,^ **	24 (22–27)
Turning around, stride, Me (IQR)	5 (5–6)	6 (5–7) *	6 (5–8)	6 (5–8) *	8 (6–9) *^,^ **	7 (6–11) *^,^ ***
Performance, yes, *n* (%)	26 (89.7)	55 (72.4) *	20 (66.7)	26 (59.1) *	12 (40) *^,^ **^,^ ***	5 (22.7) *^,^ **^,^ ***
Not performed, cognitive probl, *n* (%)	-	2 (2.6)	3 (10)	6 (13.6)	4 (13.3)	7 (31.8)
Not performed, physical probl, *n* (%)	3 (10.3)	19 (25)	7 (23.3)	12 (27.3)	14 (46.7)	10 (45.5)
Dual task cost, Me (IQR)	11.7 (0–21.7)	9.8 (−3.4–27.8)	8.8 (1.6–28.7)	11.6 (1.4–23.7)	14.7 (3.1–31.3)	16.6 (11.1–22.2)
Cost >20%, *n* (%)	7 (26.9)	18 (32.7)	6 (30)	10 (38.5)	5 (41.7)	2 (40)

*—significant when compared to normal, **—significant when compared to depression, ***—significant when compared to MCI; χ^2^ test or Fisher exact test, as appropriate, for categorical variables; Mann–Whitney test for interval variables—assessed for pairs; Abbreviations: IQR—interquartile range; Me—median value; *n*—number of cases; TUG—Timed Up and Go, TUG CB7—Timed Up and Go Counting Backward for 7, TUG EA—Timed Up and Go Enumerating Animals.

**Table 3 jcm-10-05358-t003:** Cognitive tests of patients with different performance of dual-task tests.

Cognitive Test	DT TUG Performed	DT TUG Not Performed (Cognitive Problems)	DT TUG Not Performed (Physical Problems)	*p*
TUG				
No. of patients	192	7	23	0.014
MMSE (Me, IQR)	24 (18–27)	12 (6–16)	20 (15–27)	
CDT (Me, IQR)	4 (1–6)	0 (0–1)	1 (0–5)	0.007
TUG CB7				
No. of patients	32	161	29	
MMSE (Me, IQR)	26 (23.5–28)	22 (16–26)	21 (16–26)	<0.001
CDT (Me, IQR)	5 (2.5–6)	3 (0–5)	3 (0–5)	0.012
TUG EA				
No. of patients	119	71	32	
MMSE (Me, IQR)	25 (22–27)	18 (12–24)	20 (14–26)	<0.001
CDT (Me, IQR)	5 (2–6)	2 (0–4)	2 (0–5)	<0.001
TUG manual				
No. of patients	143	19	60	
MMSE (Me, IQR)	24 (20–27)	17 (10–21)	20 (14.5–26)	<0.001
CDT (Me, IQR)	4 (2–6)	0 (0–3)	2 (0–5)	<0.001

Kruskal–Wallis range test; Abbreviations: IQR—interquartile range; Me—median value; *n*—number of cases; MMSE—Mini Mental State Examination, CDT—Clock Drawing Test, TUG—Timed Up and Go, TUG CB7—Timed Up and Go Counting Backward for 7, TUG EA—Timed Up and Go Enumerating Animals.

**Table 4 jcm-10-05358-t004:** Correlation between cognitive test score and dual task test measurements.

	TUG CB7	Stride CB7	Turn CB7	TUG NA	Stride NA	Turn NA	TUG M	Stride M	Turn M
TUG CB7	-								
Stride CB7	**0.2**	**-**							
Turn CB7	**0.75**	**0.78**	-						
TUG EA	**0.86**	**0.83**	**0.69**	**-**					
Stride EA	**0.88**	**0.91**	**0.73**	**0.79**	**-**				
Turn EA	**0.69**	**0.68**	**0.76**	**0.75**	**0.75**	**-**			
TUG M	**0.77**	**0.71**	**0.58**	**0.84**	**0.68**	**0.58**	**-**		
Stride M	**0.79**	**0.84**	**0.50**	**0.71**	**0.87**	**0.57**	**0.80**	**-**	
Turn M	**0.54**	**0.59**	**0.49**	**0.59**	**0.61**	**0.64**	**0.72**	**0.70**	**-**
MMSE	−0.16	−0.21	0.01	**−0.24**	**−0.20**	**−0.26**	**−0.23**	**−0.21**	**−0.23**

Spearman’s rank correlation coefficient-significant for *p* < 0.05 (in bold); Abbreviations: MMSE—Mini Mental State Examination, TUG—Timed Up and Go, Stride—stride number, Turn—turning around, CB7—Counting Backward for 7, EA—Enumerating Animals, M—manual.

## Data Availability

The data supporting the results in the current study are available from the corresponding author on reasonable request.

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
