# Peer review of "The Performance of Dual-Task Tests Can Be a Combined Neuro-Psychological and Motor Marker of Mild Cognitive Impairment, Depression and Dementia in Geriatric Patients—A Cross-Sectional Study"

_jcm, 2021, doi:10.3390/jcm10225358_

Round 1

Reviewer 1 Report

Introduction:

This is a cross sectional single center study conducted in sample of geriatric patients.

The aim of the study is to assess the performance of a dual task test in geriatric patients and to demonstrate an association with cognitive status.

Methods:

Patients included had a comprehensive geriatric assessment. The diagnosis of dementia was confirmed by a neuropsychological examination, which is the gold standard for diagnosis of dementia irrespective of test results..

Different dual task procedures were conducted in the same sample. Procedures were based on the timed up and go test and were combined with an additional task (CB 7, naming of animals, holding a cup).

The test "naming of animals" should be described more precisely. Do you mean enumerate animals? Then it would be a test of executive function. Or did you present pictures with animals to the patients and the animals had to be named? Then vision must be assessed as well.

Did you include subjects consecutively? 

Statistical analysis:

First, please give a calculation of sample size. Such a calculation is mandatory. Furthermore, please calculate the power of your results. Otherwise results cannot be interpreted correctly.

What is the distribution of the results? Did you observe a ceiling or floor effect in your results? I would expect this since a huge number of participants did not preform single tasks. 

Furthermore, please compare the differences of the results and divide them by standard deviation. This will give you an assessment of the relevance of your results.

Please try to conduct a ROC analysis and find a cut off value that allows for separation of subjects with and without dementia. Such a cut off value would be of useful for every day practice.

In addition, you should then calculate sensitivity, specificity, positive, and negative predictive values and correctness as well.

Please calculate the power of your results.

Of note, since you conduct multiple testing in the same sample, you must adjust the level of significance according to Bonferroni.

Results:

Table 4: please add the number of participants of each group.

Discussion:

limitations: not all geriatric patients are able to preform the TUG test. Because gait disturbances are one reason to refer patients to a geriatric unit, such gait disturbances preclude the performance of the TUG test. Therefore, results can not be generalized. Therefore, the test may be valuable only for a subgroup of patients, for example in subjects who are able to walk at all. Therefore, the additional benefit of such a dual task in geriatrics should be explained. Discuss this issue in more detail please. 

Some spelling mistakes occur in the manuscript and should be removed.

Author Response

We thank Reviewer for valuable comments. We attache response in word file.

Reviewer 2 Report

Performance of dual task tests as biomarker of mild cognitive impairment, depression and dementia in geriatric patients – a cross-sectional study.

 This is a cross-sectional cohort study investigating dual task motor performance and its possible association with cognitive impairment (MCI to dementia) and depressive symptomatology in older adults.  In general, the manuscript should be of interest to a wide range of health and allied health professionals who are currently involved in the care and management of older adults. The manuscript unfortunately would need careful editing to address the numerous typographical and grammatical errors throughout the submission.

I have outlined some queries and suggestions that the authors may wish to consider.

Title:

The investigation is really looking at neuropsychological and motors performance, rather than what most physician would consider as biological markers. It would be helpful if the authors could revise the title to reflect this.

Abstract:

All of the abbreviations should be spelled out rather than readers have to search the manuscript for their meaning i.e., MCI, TUG CB7, NA etc.

The abstract suffers from poor grammar and stylistic errors.

Introduction:

Generally, well written, but terse on previous empirical evidence in this area. and lacks a discussion on TUG, dual tasks and executive function.    

Methods:

It is unclear to me how the diagnosis of cognitive impairment (MCI, dementia) or depression was reached. A diagnosis of MCI or dementia cannot made alone on the results of neuropsychological screening or pencil and paper tests for mood disorders. At the very least, the application of for example DSM-5 criteria and who and how this decision was reached would need to be stated.

Whilst the TUG is a validated motor assessment, I do not believe that the TUG CB7, NA, or manual are validated. If this is the case, the authors would need to provide some validated evidence of these assessments reliability and validation.  

Results:

The authors should have considered employing a general linier model or MANOVA when comparing multiple groups within the cohort.

The results of the various TUG are adequately described in the results section. However, Table 3 is basically repeating what is already given in the text and it might be better to collapse the different cognitive states into normal, MCI and dementia for ease of reading and interpretation.

Discussion:

The discussion is in general well written and outlines the findings of the study. However, the biggest weakness is that the authors do not describe the relationship with dual task performance and the executive function. The TUG should have been assessed against a ‘Gold Standards’ of executive function instrument such as the Trail Making Tests or Stroop test. Without this, the study only tells us that individuals with and without cognitive impairment and depressive symptoms on dual task assessments.   

Other:

Table 1: Should be incorporated in to the methods section or added as a supplemental table.

Figure 1: Should be removed and the cognitive state reported as a table or as text.

Figure 3: Should be remove and reported alone in the text

Table 5: Discuss the fact that although the MMSE was significantly correlated, the correlations were very weak and I assume that the CDT correlations were similar.

Author Response

(The authors gave the same response as above.)

Round 2

Reviewer 1 Report

comments have been adressed now.

Please see former evaluation of the manuscript.

Reviewer 2 Report

Thank you for letting me review the revised manuscript. The authors have clearly taken on board the comments raised by myself and the other reviewer and have made significant changes to the manuscript as requested. 

I am happy to report that the authors have addressed the concerns that I raised with the original manuscript.